# Cell-Extrinsic Differentiation Block Mediated by EphA3 in Pre-Leukaemic Thymus Contributes to Disease Progression

**DOI:** 10.3390/cancers13153858

**Published:** 2021-07-31

**Authors:** Adriana C. Pliego Zamora, Hansini Ranasinghe, Jessica E. Lisle, Chun Ki Ng, Stephen Huang, Racheal Wadlow, Andrew M. Scott, Andrew W. Boyd, Christopher I. Slape

**Affiliations:** 1The University of Queensland Diamantina Institute, The University of Queensland, Brisbane 4102, Australia; a.pliegozamora@uq.edu.au (A.C.P.Z.); hansini.ranasinghe@uq.net.au (H.R.); j.lisle@uq.edu.au (J.E.L.); chun.ng@uq.edu.au (C.K.N.); s.huang@uq.edu.au (S.H.); r.wadlow@uq.edu.au (R.W.); 2Olivia Newton-John Cancer Research Institute and La Trobe University, Heidelberg 3084, Australia; andrew.scott@onjcri.org.au; 3Faculty of Medicine, University of Melbourne, Melbourne 3000, Australia; 4Department of Medicine, The University of Queensland, Brisbane 4072, Australia; aw.boyd@uq.edu.au

**Keywords:** T-ALL, Eph, pre-leukaemia, cell competition

## Abstract

**Simple Summary:**

The *NUP98-HOXD13* (*NHD13*) mouse is a model of T-cell leukaemia (T-ALL) featuring a pre-leukemic phase, in which T-cell progenitors from the thymus of an *NHD13* mouse can engraft into the thymus of a recipient mouse—an ability that normal T-cell progenitors do not possess. However, loss of this engraftment ability (by deletion of the *Lyl1* gene) did not result in any loss of leukemogenesis activity, indicating the activity of redundant oncogenic pathways in this model. Having observed an overexpression of the EphA3 protein in the *NHD13* thymocytes, we hypothesized that this gene might be involved in a redundant leukaemogenic pathway. Deletion of *EphA3* did not affect the engraftment ability of the thymocytes, but did reduce the incidence of T-ALL. We thus uncovered a distinct mechanism of leukaemogenesis, which we believe operates in parallel to that mediated by *Lyl1*.

**Abstract:**

We recently characterised the *NUP98-HOXD13* (*NHD13*) mouse as a model of T-cell pre-leukaemia, featuring thymocytes that can engraft in recipient animals and progress to T-cell acute lymphoblastic leukaemia (T-ALL). However, loss of this engraftment ability by deletion of *Lyl1* did not result in any loss of leukemogenesis activity. In the present study, we observe that *NHD13* thymocytes overexpress EPHA3, and we characterise thymocyte behaviour in *NHD13* mice with deletion of *EphA3*, which show a markedly reduced incidence of T-ALL. Deletion of *EphA3* from the *NHD13* mice does not prevent the abnormal accumulation or transplantation ability of these thymocytes. However, upon transplantation, these cells are unable to block the normal progression of recipient wild type (WT) progenitor cells through the normal developmental pathway. This is in contrast to the *EphA3*^+/+^ *NHD13* thymocytes, which block the progression of incoming WT progenitors past the DN1 stage. Therefore, *EphA3* is not critical for classical self-renewal, but is essential for mediating an interaction between the abnormally self-renewing cells and healthy progenitors—an interaction that results in a failure of the healthy cells to differentiate normally. We speculate that this may orchestrate a loss of healthy cell competition, which in itself has been demonstrated to be oncogenic, and that this may explain the decrease in T-ALL incidence in the absence of *EphA3*. We suggest that pre-leukaemic self-renewal in this model is a complex interplay of cell-intrinsic and -extrinsic factors, and that multiple redundant pathways to leukaemogenesis are active.

## 1. Introduction

T-cell acute lymphoblastic leukaemia (T-ALL) is a clonal malignancy caused by the accumulation of genomic lesions that disrupt the development of T cells [1,2,3,4]. T-ALL does not present clinically until it is in its acute phase, but there is evidence from both murine models and humans that a pre-malignant phase exists [5,6,7]. Understanding the drivers of this pre-malignant phase, and the drivers of subsequent transformation to acute phase disease, may identify new modes of clinical surveillance and therapy.

The *NUP98*-*HOXD13* (*NHD13*) transgenic mouse is a model of HoxA-driven T-ALL, with pre-leukaemic thymocytes that exhibit abnormal behaviour prior to the development of overt T-ALL. *NHD13* thymocytes accumulate abnormally in the DN2 differentiation stage, and are capable of engraftment upon transplantation. After a latency period, the mice develop T-ALL [6,8,9]. *NHD13* thymocytes require the *Lyl1* gene for their engraftment ability, but although *NHD13 Lyl1* knockout (*NHD13*-*Lyl1KO*) thymocytes cannot engraft, *NHD13*-*Lyl1KO* mice still succumb to T-ALL at the same rate [6]. Therefore, it appears that the abnormal self-renewal ability of *NHD13* thymocytes—for which engraftment is regarded as a proxy—is not required for the induction of T-ALL. This stands in contrast to the *Lck*-*LMO2* transgenic mouse model of LMO-driven T-ALL, in which abnormal self-renewal of thymocytes is also dependent on *Lyl1*, but deletion of *Lyl1* prevents the formation of T-ALL [7]. This suggests a redundancy of the oncogenic mechanisms in the *NHD13* mice that may be specific to the HoxA-driven T-ALLs, and led us to investigate what other mechanisms may be occurring.

We noted abnormal expression of EPHA3 in the expanded DN2 population of NHD13 thymocytes. *EphA3* is a member of the Eph family of cell-surface receptor tyrosine kinases which, with their ephrin ligands, can modulate cell adhesive properties as well as coordinate cell movement, and play critical roles in development, tissue homeostasis, and cancer [10,11,12,13,14]. Importantly, *EphA2* (which is closely related to *EphA3*) is important for tumour-suppressive cell competition in epithelial sheets [15]. Our focus—*EphA3*—is overexpressed in different types of human leukaemia, including T-ALL [16,17,18,19]. There is evidence implicating *EphA3* in regulating stem-like features in leukaemic stem cells (LSCs) [20] and solid cancer stem cells (CSCs) [21,22]. EPHA3 is a promising therapeutic target in cancer [23], including in glioblastoma [24,25]. Moreover, EPHA3 is a viable anti-leukaemic target, with the success of an activating monoclonal antibody (IIIA4), targeted payload delivery, or RNAi-mediated EPHA3 knockdown in models of multiple myeloma [26] and pre-B-ALL [27], as well as in human myeloid malignancies [28]. We hypothesized that *EphA3* might be involved in a parallel oncogenic pathway, and investigated its involvement in the *NHD13* T-ALL disease.

## 2. Materials and Methods

### 2.1. Mouse Strains

Transgenic *NHD13* mice [9] and *EphA3* germline knockout (*EphA3KO*) mice [29] have been previously described. The double-transgenic *NHD13*-*EphA3*^−/−^ mouse was generated by crossing *NHD13* and *EphA3KO* mice. All strains were back-crossed at least 12 generations onto a C57BL/6 background. CD45.1 WT mice were purchased from ARC, Perth, Australia. Mice were euthanised at 6 or 12 weeks of age accordingly, using CO_2_, and then thymuses were dissected. Animal experiments were approved by the Animal Ethics Committee of the University of Queensland.

### 2.2. Tissue Processing

Single-cell thymocyte suspensions were obtained by gently pressing the thymus onto a 40-µm cell strainer (BD Falcon, Franklin Lakes, NJ, USA) with a syringe plunger and rinsing with PBS supplemented with 2% FBS (2% FBS–PBS). Cell counts and cell viability were determined using Trypan blue. Thymocyte samples were used for flow cytometry analysis, transplantation experiments, or stored in TRIzol (Invitrogen, Waltham, MA, USA) at −80°C.

### 2.3. Flow Cytometry: Thymocyte DN Populations, Chimerism, and Cell Cycle

For each mouse, 1 × 10^6^ thymocytes were blocked with anti-CD16/32 (FcR receptor) and stained with Zombie Aqua, lineage (B220, TER119, GR-1, MAC-1), CD4, CD8, CD44, CD25, C-KIT, and EPHA3. Double-negative (DN) populations were determined as lineage^-^, CD4^−^, CD8^−^, CD44^−^^/+^, CD25^−^^/+^, and C-KIT^−^^/+^. The expression of EphA3 was evaluated in populations DN 1–4. In addition to the DN panel, CD45.1 and CD45.2 were included for chimerism studies. For cell cycle studies, cells were stained as previously described for extracellular markers, then fixed and permeabilized with Fix & Perm Cell Permeabilization reagents (Thermo Fisher, Waltham, MA, USA), stained first with Ki67, and then with 20 µM Hoechst 33342 (Sigma-Aldrich/Merck, Burlington, VT, USA), in a solution with 0.05% saponin (Sigma-Aldrich/Merck) and 1 µg/mL RNase A (Sigma-Aldrich/Merck) [30]. Cells were analysed by flow cytometry using a BD LSR FortessaX-20. Data were analysed using FlowJo 10.

### 2.4. Thymocyte Transplantation

Six-week-old donors (CD45.2) of each indicated genotype were euthanised using CO_2_ and their thymuses were dissected. Single thymocyte suspensions were obtained as previously described. One-sixth of the thymus in 200 µL of PBS was intravenously injected into female mice (CD45.1) sub-lethally irradiated with 6.5 Gy from a ^60^C γ. Four weeks post-transplantation, recipients were euthanised and a serial transplantation was performed as described above, for a total of 4 subsequent times. Samples of thymocytes were stained and analysed for chimerism and cell cycle studies, as previously described.

### 2.5. Gene Expression Study

Total RNA was isolated from thymocytes using TRIzol (Invitrogen) reagent and a GenElute mammalian total RNA miniprep kit (Sigma-Aldrich). Day 17 embryonic RNA was used to validate primers (Clontech/Takara, Kyoto, Japan). cDNA was synthesised with Superscript III (Invitrogen). Primers for *HoxA5*, *HoxA7*, *HoxA9*, *HoxA10*, *Lmo2*, *Hhex*, and *Klf2* were designed in-house and purchased from IDT. For all qRT-PCR reactions, PowerUp™ SYBR™ Green Master Mix (Applied Biosystems, Waltham, MA, USA) was used, except for *Lyl1*, for which TaqMan Fast Advanced Master Mix (Applied Biosystems) was used. Relative expression of target transcripts was analysed using the ViiA 7 Real-Time PCR System (Thermo Fisher Scientific, Waltham, MA, USA). Expression was normalised to two reference genes: *Hprt1* and *B2m*.

### 2.6. Disease Progression Study

Groups of mice were designed to enable detection of a relative hazard of 0.5 with type I and II error rates of 0.3. Mice that reached ethical endpoints were euthanised using CO_2_ and their thymomas/thymuses were harvested. T-ALL cell profiles were used to corroborate diagnosis. Single thymocyte samples were obtained, stained, and analysed via flow cytometry, as previously described.

### 2.7. Statistical Analyses

GraphPad Prism 7.3. was used for analysis. We used one-way ANOVA with Tukey’s post hoc analysis. Survival curve analysis was carried out using the Mantel–Cox test. * *p* < 0.03, ** *p* < 0.002, *** *p* < 0.0002, and **** *p* < 0.0001. Bars represent the SEM.

## 3. Results

### 3.1. EphA3 Is Abnormally Expressed in NHD13 Thymocytes

We characterised WT and *NHD13* thymocytes by flow cytometry (Figure 1A). Consistent with previous studies [6,8], the *NHD13* thymuses showed an abnormal accumulation of thymocytes in the DN2A (CD44^hi^, CD25^+^) stage (Figure 1B). We also observed an abnormally high level of expression of the C-KIT protein in the CD25^+^ fraction (Figure 1B), as previously described [6]. We examined the expression of EPHA3 via flow cytometry, and found that EPHA3 is specifically expressed in a subset of the expanded DN2A (CD4^−^, CD8^−^, CD44^+^, CD25^+^) population of *NHD13* thymocytes (Figure 1C).

### 3.2. EphA3 Deletion Delays the NHD13 Thymic Phenotype

To assess the role of *EphA3* in the *NHD13* thymus, we crossed the *NHD13* transgene onto an *EphA3* germline deletion background to generate *EphA3*-deficient *NHD13* mice (*NHD13*-*EphA3*^−^^/^^−^). We confirmed the absence of EPHA3 in the thymocytes via flow cytometry (Figure 2A). Analysis of the thymuses of each genotype (WT, *EphA3*^−^^/^^−^, *NHD13*-*EphA3*^+/+^, and *NHD13*-*EphA3*^−^^/^^−^) at 6 weeks of age showed that the CD4/CD8 cell profiles (Figure 2B and Appendix A) were similar, and that the overexpression of C-KIT remained high in both *NHD13*-*EphA3*^+/+^ and *NHD13*-*EphA3*^−^^/^^−^ DN2A thymocytes (Figure 2B). However, there were significant differences in the cellularity of the thymus and the proportions of the DN subpopulations (CD44/CD25). The reduced cellularity of the *NHD13* thymus was corrected by the deletion of *EphA3* (Figure 2D). Furthermore, the accumulation of DN2A cells in the *NHD13* thymus was decreased by the deletion of *EphA3* (Figure 2B,D). The proportions of the DN1 and DN2B populations in the *NHD13*-*EphA3*^−^^/^^−^ thymus were similar to those in the WT and *EphA3*^−^^/^^−^ control thymuses, correcting the abnormalities seen in the *NHD13*-*EphA3*^+/+^ thymus. The proportion of DN2A and DN3 cells was also partially normalised (Figure 2D). The incomplete nature of these corrections, however, suggested that a biological difference between the *NHD13*-*EphA3*^−^^/^^−^ and the WT and *EphA3*^−^^/^^−^ controls remains.

To gain further insight, we characterised a cohort of older mice. At 12 weeks of age, the loss of *EphA3* did not alter the phenotype of *NHD13* thymuses, as shown by the similarity in CD4/CD8 cell profiles (Figure 2C), with significantly increased DN, SP4, and SP8 populations as well as decreased DP populations (Appendix A) in both the *NHD13*-*EphA3*^+/+^ and *NHD13*-*EphA3*^−^^/^^−^ thymuses. Moreover, the low cellularity (Figure 2E), accumulation of DN2A thymocytes (Figure 2C,E), and increased DN1 and decreased DN3 proportions (Figure 2E) were the same in both *NHD13*-*EphA3*^+/+^ and *NHD13*-*EphA3*^−^^/^^−^ thymuses. Therefore, phenotypes that were rescued or partially rescued at 6 weeks of age were once again abnormal at 12 weeks of age.

In the thymus of WT mice, the loss of *EphA3* had no effect at either timepoint, exhibiting no significant changes in the thymus cellularity (Figure 2D,E), equal proportions of DN, DP, SP4, SP8 (Appendix A), and DN1-4 (Figure 2D,E) populations, and equal cell profiles (Figure 2B,C) in both WT and *EphA3*^−^^/^^−^.

Taken together, these results suggest that the deletion of *EphA3* delays, but does not prevent, the NHD13 thymus phenotype.

### 3.3. EphA3 Deletion Reduces T-ALL Incidence in NHD13 Mice

To directly determine the impact of the absence of *EphA3* on T-ALL occurrence in this model, we compared the T-ALL-free survival of *NHD13*-*EphA3*^+/+^ and *NHD13*-*EphA3*^−^^/^^−^ mice (Figure 3A). Both wild-type (*n* = 13) and *EphA3*^−^^/^^−^ (*n* = 17) cohorts survived without the development of any T-ALL. T-ALL occurred in the *NHD13*-*EphA3*^−^^/^^−^ mice at a lower frequency than in the *NHD13*- *EphA3*^+/+^ mice (9/30 (30%) *NHD13*-*EphA3*^+/+^ mice developed T-ALL, as opposed to 4/25 (16%) *NHD13*-*EphA3*^−^^/^^−^ mice). The T-ALL-free survival curves obtained with the Mantel–Cox test were significantly different (*p* = 0.0418). The curves overlap early in the disease course and then deviate late, perhaps suggesting that *EphA3* is especially important in late-onset T-ALL in this model. It is important to note that NHD13 mice also develop acute myeloid leukaemia (AML), separate from the development of T-ALL in the thymus^6^. There was no difference between *NHD13*-*EphA3*^−^^/^^−^ mice and *NHD13*- *EphA3*^+/+^ mice in the incidence of AML in this cohort (Appendix A), and AML events were censored from the T-ALL-free survival curve. The only difference seen between *NHD13*-*EphA3*^−^^/^^−^ and *NHD13*- *EphA3*^+/+^ survival was due to the reduced incidence of T-ALL.

We compared the immunophenotypes of several T-ALLs arising from *NHD13*-*EphA3*^+/+^ or *NHD13*-*EphA3*^−^^/^^−^ mice (Figure 3B,C). No differences were noted in the CD4/CD8 profiles between groups, as they vary considerably from sample to sample, independently of the presence of *EphA3*. The DN (CD44/CD25) profiles from the *NHD13*-*EphA3*^+/+^ mice show an accumulation of thymocytes in the DN1 (4 of 5) and DN2A (3 of 5) populations (Figure 3B), whereas in the absence of *EphA3*, the DN (CD44/CD25) profiles show thymocytes in differentiation stages other than DN1, including DN3 and DN4. Notably, only one of the five *EphA3*^+/+^ T-ALLs demonstrated appreciable EPHA3 expression. Together, this demonstrated that deletion of *EphA3* reduced—but did not eliminate—the incidence of T-ALL in the *NHD13* model.

### 3.4. Sustained Abnormal Expression of Self-Renewal Genes in the Absence of EphA3

The stem-like transcriptional signature downstream of the *NUP98*-*HOXD13* fusion protein has *LMO2*-like, *Lyl1*-dependent, and *Lyl1*-independent components [6]. To determine any impact of *EphA3* deletion on this self-renewal signature, we compared the expression of several candidate self-renewal regulatory genes in *NHD13*-*EphA3*^+/+^ and *NHD13*-*EphA3*^−^^/^^−^ thymocytes, using qRT-PCR. We previously showed overexpression of HoxA genes such as *HoxA5, HoxA7, HoxA9*, and *HoxA10* in *NHD13* DN thymocytes [8]. We found no differences in the profiles of overexpression of these genes in whole *NHD13* thymus tissue in the presence or absence of *EphA3* (Figure 4). Similarly, oncogenes such as *Lmo2, Lyl1, Hhex*, and *Klf2*—which are all deregulated in *NHD13* DN2 thymocytes—showed no significant changes in expression when comparing *NHD13*-*EphA3*^+/+^ and *NHD13*-*EphA3*^−^^/^^−^ DN2 thymocytes (Figure 4). These results suggest that *EphA3* exerts influence via an independent mechanism to those previously described in T-ALL mouse models.

### 3.5. EphA3 Mediates an Interaction between Donor and Recipient Thymocytes

The *NHD13* thymocytes’ abnormal ability to self-renew is functionally defined by their capacity to engraft upon transplantation, and to retain this ability indefinitely upon serial transplantation.

To assess the engraftment ability of *NHD13* thymocytes in the absence of *EphA3*, we performed serial transplantation of *NHD13*-*EphA3*^−^^/^^−^ thymocytes into sub-lethally irradiated wild-type mice, and evaluated donor chimerism 4 weeks later (Figure 5A). We found that *NHD13*-*EphA3*^−^^/^^−^ thymocytes are able to engraft similarly to *NHD13*-*EphA3*^+/+^ thymocytes (Figure 5B), with a donor contribution in the DN compartment of 70% and 65% in the first and second transplants, respectively (Figure 5C). Strikingly, the *NHD13*-*EphA3*^−^^/^^−^ thymocytes failed to engraft upon the third serial transplant, whereas the *NHD13*-*EphA3*^+/+^ thymocytes engrafted indefinitely (Figure 5B,C). In parallel, the cellularity of the thymus (Figure 5D) and the proportions of the DN1-4 populations (Figure 5E) were rescued by the third transplant in the *NHD13*-*EphA3*^−^^/^^−^ recipients, and were similar to the WT-*EphA3*^+/+^ and WT-*EphA3*^−^^/^^−^ control recipients. These experiments were performed on three occasions from independent donors, and similar results were seen in each replicate. Therefore, *EphA3* is dispensable for primary and secondary engraftment, but essential for serial engraftment.

Furthermore, in the primary transplants that received *NHD13*-*EphA3*^−^^/^^−^ thymocytes, the recipient WT thymocytes (CD45.1) exhibited a strikingly different DN cell profile compared to the thymuses transplanted with *NHD13*-*EphA3*^+/+^ thymocytes (Figure 5B). In the *NHD13*-*EphA3*^+/+^ primary recipient, there was an accumulation of the recipient thymocytes in the DN1 stage compared to either the WT or *EphA3*^−^^/^^−^ thymuses (Figure 5B). These cells appear to be prevented from differentiating beyond the DN1 stage, despite being genetically wild type. However, despite a similar level of engraftment of donor cells in the *NHD13*-*EphA3*^−^^/^^−^ recipient, the WT recipient cells (CD45.1) show a relatively normal DN distribution (Figure 5F). This suggests that, upon transplantation of *NHD13* thymocytes, *EphA3* is required to prevent normal differentiation of the incoming WT progenitor cells.

### 3.6. NHD13 DN2 Thymocytes Induce Cell Cycle Arrest on WT Cells Independently of EphA3 Expression

Another characteristic of *NHD13*-*EphA3*^+/+^ thymocytes is their increased quiescence compared to WT cells [6]. This has not yet been assessed in transplant recipients, nor in the context of *EphA3*. To investigate whether EPHA3 expression modulates the cell division of *NHD13* thymocytes and neighbouring wild-type thymocytes, we analysed the cell cycling status of DN2 (DN2A and DN2B) thymocytes in the primary transplant recipients.

We found that, following transplant, donor *NHD13*-*EphA3*^+/+^ thymocytes are more quiescent than recipient thymocytes from control WT-*EphA3*^+/+^ recipient mice (in which no donor cells engraft) (Figure 6). This is similar to what was previously described for *NHD13* thymocytes prior to transplantation [6]. However, there were no differences in the cell cycle status in the *NHD13*-*EphA3*^−^^/^^−^ cells compared to the *NHD13*-*EphA3*^+/+^ cells (Figure 6), indicating that *EphA3* deletion does not affect the cell cycle status of *NHD13* thymocytes. We also observed that in both *NHD13*-*EphA3*^+/+^ and *NHD13*-*EphA3*^−^^/^^−^ transplants, the WT recipient cells (CD45.1) divided at a lower rate compared to the equivalent WT recipient cells (CD45.1) in the thymuses transplanted with WT-*EphA3*^+/+^ or WT-*EphA3*^−^^/^^−^ thymocytes (Figure 6). This observation suggests that, independently of EphA3 expression, *NHD13* thymocytes hamper the cell division of proximal WT thymocytes. We also analysed the cell cycle status of the DN1, DN3, and DN4 thymocytes (Appendix A) and found that, although *NHD13* cells generally cycle slower than wild-type cells, the effect of the *NHD13* thymocytes on the wild-type recipients only occurs in DN2.

Therefore, we conclude that multiple cell-extrinsic effects are perpetrated on wild-type recipient thymocytes by *NHD13* thymocytes in the transplant setting. Among these, the block in differentiation of recipient thymocytes is dependent on the expression of EPHA3 by *NHD13* thymocytes, and prevention of this effect via the deletion of *EphA3* results in a reduction in the incidence of T-ALL in the *NHD13* model.

## 4. Discussion

Pre-leukaemic stem cells have previously been characterised on the basis of their ability to self-renew inappropriately, as defined by an engraftment assay. This has been considered a cell-intrinsic ability, granted by the inappropriate expression of stem-like transcription factors. However, the removal of this transplantation ability from *NHD13* thymocytes, via the deletion of *Lyl1*, does not impact on T-ALL incidence [6]. Therefore, there must be another, perhaps redundant, mechanism responsible.

In the present study, we characterise the role of *EphA3* in pre-leukaemic thymocytes in the *NHD13* mouse model. EPHA3 is typically considered to be undetectable in adult tissues (both mouse and human), and has been described extensively as being overexpressed in a wide variety of haematological malignancies [19,26,27,28]. In myeloid leukaemias and other cancers, *EphA3* has been identified as a regulator of stem cell abilities [20,21,22,31]. We find that, in the *NHD13* mouse model, *EphA3* expression in the thymus is restricted to the expanded DN2 compartment. We show that, although *EphA3* deletion delays the phenotype, *EphA3* is not required for the NHD13 thymic phenotype. NHD13 thymuses have an expanded DN2 compartment, increased DN1 compartment, and reduced overall cellularity. The expanded population displays a marked overexpression of C-KIT, which is important in early T-cell differentiation in the thymus, and which is required for the transplantation ability and radioresistance of *Lmo2* transgenic thymocytes. [32,33,34,35]. *EphA3* deletion reduces each of these, to an extent, at 6 weeks of age, but makes no difference to these outcomes in 12-week-old mice.

In a survival study, however, *EphA3* deletion reduced the incidence of T-ALL in *NHD13* mice, decreasing the incidence by almost half (30% *NHD13*-*EphA3*^+/+^ compared to 16% *NHD13*-*EphA3*^−^^/^^−^ mice). Much of the difference is seen in older mice, perhaps suggesting that *EphA3* is especially important in late-onset T-ALL in this model. We investigated the expression of several genes believed to be involved in the abnormal self-renewal and leukaemogenicity of these cells, and found that all continued to be abnormally overexpressed in *NHD13* thymocytes in the absence of *EphA3*.

Our transplantation study showed that, although *EphA3* is not required for the engraftment capacity of pre-leukaemic *NHD13* thymocytes, there is an important difference in the behaviour of the wild-type recipient thymocytes in this setting. When transplanted with *NHD13* thymocytes, the recipient animal’s own thymocytes are unable to progress beyond DN1, seemingly blocked by the expanded DN2 population of *NHD13* donor cells. However, when *EphA3* is deleted from the *NHD13* donor cells, the recipient thymocytes are able to progress normally through DN2 and DN3, to DN4 and beyond.

Previous studies have demonstrated that continual import of progenitors from the bone marrow into the thymus is essential to maintain cell competition in the thymus—a key process to sustain the normal differentiation and homeostasis of thymocytes. The absence of normal cell competition results in the abnormal self-renewal of WT thymocytes, and is leukaemogenic [36,37,38,39,40]. In the models used in the studies that demonstrated this, the recipient mice were deficient in incoming progenitors, so the incumbent thymocytes were not outcompeted, and could instead remain in the thymus. This long occupancy presumably permits them time to accumulate mutations and become leukaemogenic. In our *NHD13* thymocyte transplantation model, following the four-week engraftment period, recipient thymuses contained two populations of cells: *NHD13* cells that were transplanted and engrafted in the thymus, and wild-type “progenitor” cells that were newly arrived from the bone marrow. At 4 weeks post-transplantation, incoming WT cells had accumulated in the DN1 stage, as they were prevented from progressing through their normal differentiation pathway by the presence of *NHD13*-*EphA3*^+/+^ DN2 cells. This may result in an inefficient cell competition in DN3, with no “young” progenitors coming in to outcompete the incumbent “old” progenitors. When *EphA3* is deleted, although transplanted *NHD13*-*EphA3*^−^^/^^−^ cells still accumulate in the DN2 stage, the incoming WT cells complete their differentiation process normally, which may restore cell competition in DN3. This difference in the progression of incoming thymic progenitors may also explain the differences seen between non-transplanted *NHD13*-*EphA3*^+/+^ and *NHD13*-*EphA3*^−^^/^^−^ thymuses at six weeks of age, but that were absent at 12 weeks of age (Figure 2). The progression of incoming progenitors beyond DN1 in the absence of *EphA3* is more normal than in the presence of *EphA3*, but because these incoming progenitors are still expressing *NHD13*, they mostly accumulate in DN2, and predominantly do not progress normally. This is different to the transplant setting, where the incoming progenitors are wild-type recipient cells, and will progress normally if they are able.

Based on the extracellular nature of *EphA3*, and the fact that we have shown that DN2 *NHD13*-*EphA3*^+/+^ thymocytes are known to be responsible for the impaired thymocyte turnover, it seems likely that EPHA3 may mediate a cell–cell interaction that stops the incoming WT thymocytes in the DN1 stage, and prevents new progenitors from being imported into the thymus. This “blockade” capacity is independent of the ability of *NHD13* thymocytes to accumulate in the DN2 stage, as this accumulation occurs in the absence of *EphA3*. This accumulation is likely driven by the maintained self-renewal gene expression signature of the *Lyl1* gene.

In the *NHD13* mouse model, we propose that the reduction of the cell competition inhibition mechanism via the deletion of *EphA3* is insufficient to prevent the development of T-ALL (although it does reduce the incidence). Cell competition is restored by deletion of *EphA3*, but the intrinsic self-renewal capacity of the *NHD13* thymocytes driven by *Lyl1* [6] remains intact. As reported previously, although *Lyl1* is required for the transplantability and establishment of a stem-cell-like gene expression program in *NHD13* thymocytes, deletion of *Lyl1* is also not sufficient to prevent T-ALL [6]. Therefore, we suggest that the development of T-ALL in the *NHD13* model is prompted by a complex interplay of cell-intrinsic and cell-extrinsic factors. It is clear that *NHD13* thymocytes employ at least two different mechanisms of leukaemogenesis, and that each is sufficient for leukaemia to occur: (1) the expression of the oncogene *Lyl1*, and (2) the inhibition of cell competition mediated by EPHA3.

Our findings also report insights into new cellular mechanisms by which *NHD13* pre-leukaemic stem cells are able to manipulate WT normal cells. We found that both *NHD13* and wild-type DN2 cells in *NHD13*-recipient thymuses divide at a slower rate. This suggests that the presence of the *NHD13* cells can inhibit the normal cell division of the wild-type DN2 cells. This effect was independent of *EphA3*, indicating that there is more than one means of communication between *NHD13* and wild-type thymocytes in our transplant model.

## 5. Conclusions

In the present study, the deletion of *EphA3* resulted in the restoration of normal cell competition in *NHD13* thymuses, and reduced the incidence of T-ALL as a result. This supports earlier work [37] demonstrating that cell competition is a tumour suppressor in the thymus, in a model that was not specifically engineered to disrupt incoming thymocyte progenitors. We have, for the first time, identified a cell competition mechanism in a non-engineered model, and further identified a molecular mediator responsible for the establishment of this mechanism. Given the perceived role of *EphA3* in self-renewal in other settings, it is interesting to ponder whether these phenomena might also be due to a contribution of cell competition to self-renewal.

## Figures and Tables

**Figure 1 cancers-13-03858-f001:**
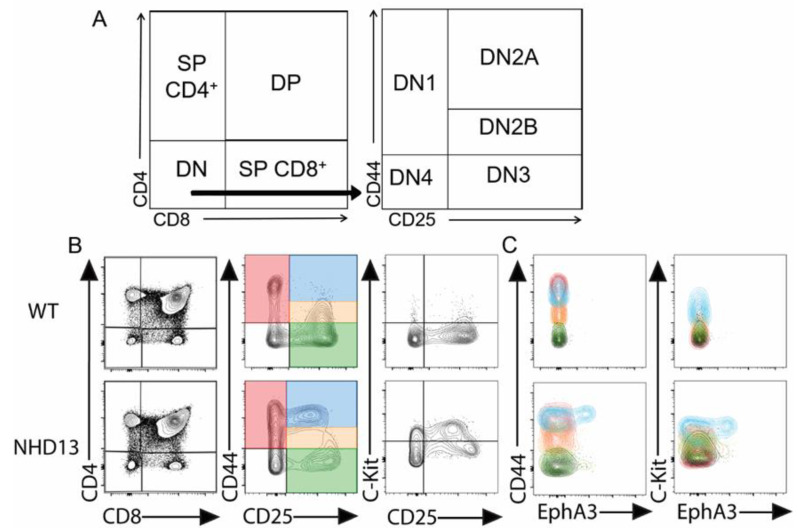
EPHA3 is expressed in DN2A *NHD13* thymocytes. (**A**) Gating strategy to examine thymocytes via flow cytometry. (**B**) Representative cell profiles using CD4/CD8, CD44/CD25, and C-KIT/CD25. (**C**) Detection of EPHA3 in DN1-4. The colour of the contours matches the colour of the gated DN1-4 populations from panel B.

**Figure 2 cancers-13-03858-f002:**
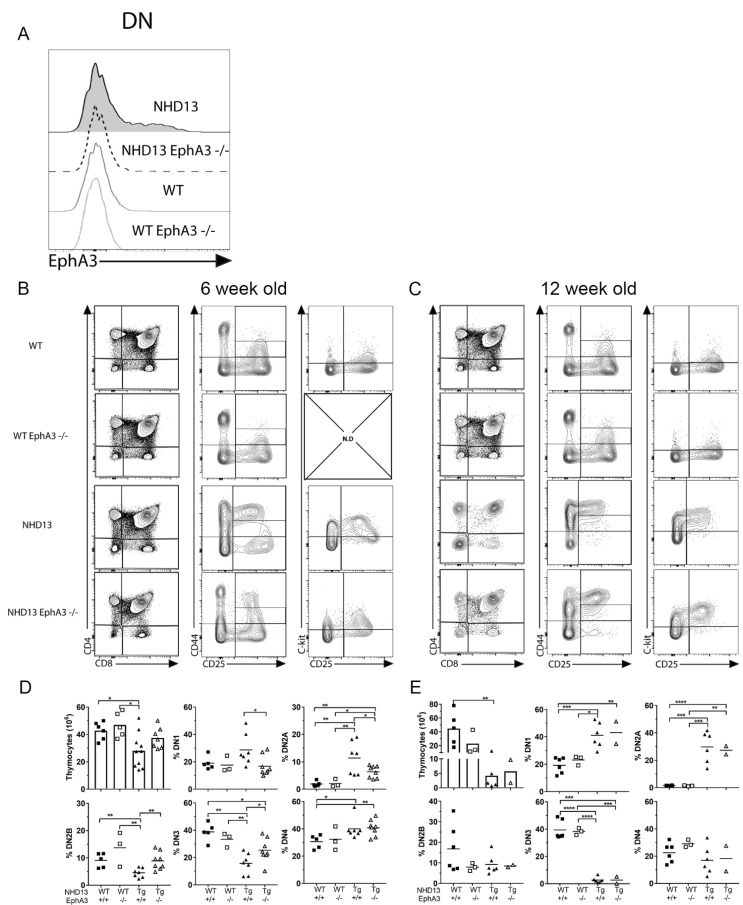
The *NHD13* thymocyte differentiation block is transiently rescued by loss of *EphA3*. (**A**) Confirmation of *EphA3* deletion in DN (CD4^−^/CD8^−^) thymocytes by flow cytometry. (**B**) Representative thymocyte profiles using CD4/CD8, CD44/CD25, and C-KIT/CD25 for each indicated genotype in 6-week-old and (**C**) 12-week-old mice. (**D**) Quantitation of thymic cellularity and proportions of DN1, DN2A, DN2B, DN3, and DN4 in 6-week-old and (**E**) 12-week-old mice. Data represent the mean, while points represent individual mice (3–8 per group). *p*-values were calculated using Student’s *t*-test (* *p* < 0.05, ** *p* < 0.005, *** *p* < 0.0005, **** *p* < 0.00005). N.D.: not done.

**Figure 3 cancers-13-03858-f003:**
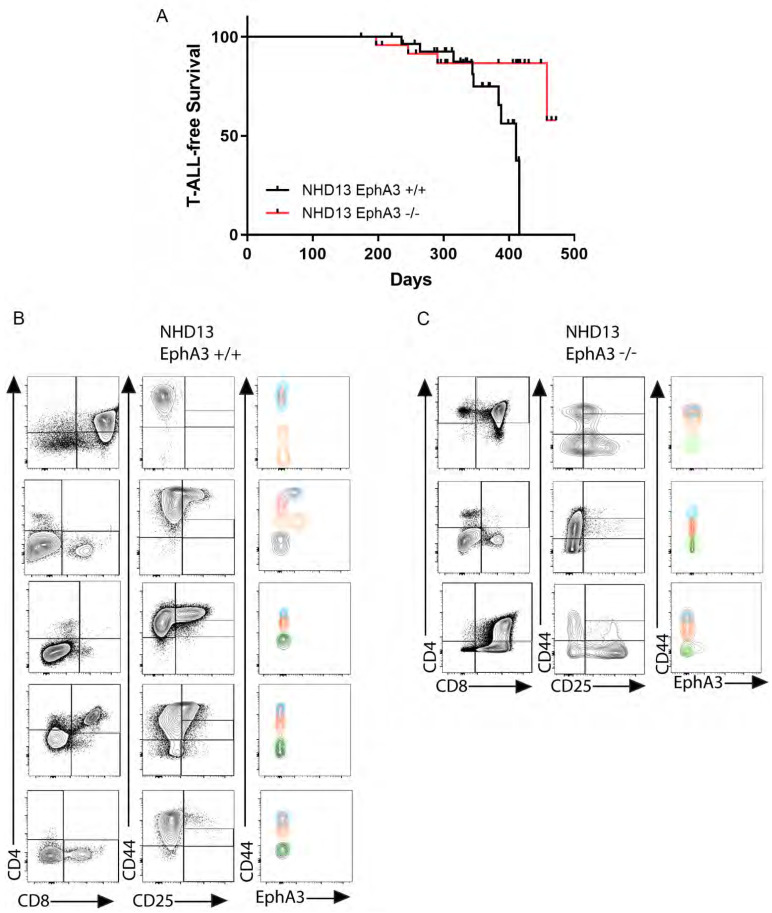
EphA3 deletion reduces T-ALL incidence in *NHD13* mice. (**A**) Kaplan–Meier plot of T-ALL-free survival for each indicated genotype, with deaths due to causes other than T-ALL excluded. (**B**,**C**) Representative cell profiles and EPHA3 detection in T-ALL leukaemias of (**B**) *NHD13 EphA3*^+/+^ and (**C**) *NHD13 EphA3*^−/−^ mice, using CD4/CD8, CD44/CD25, and CD44/EPHA3. The colour of the contour plots matches the colour of the gated DN1-4 populations from Figure 1B.

**Figure 4 cancers-13-03858-f004:**
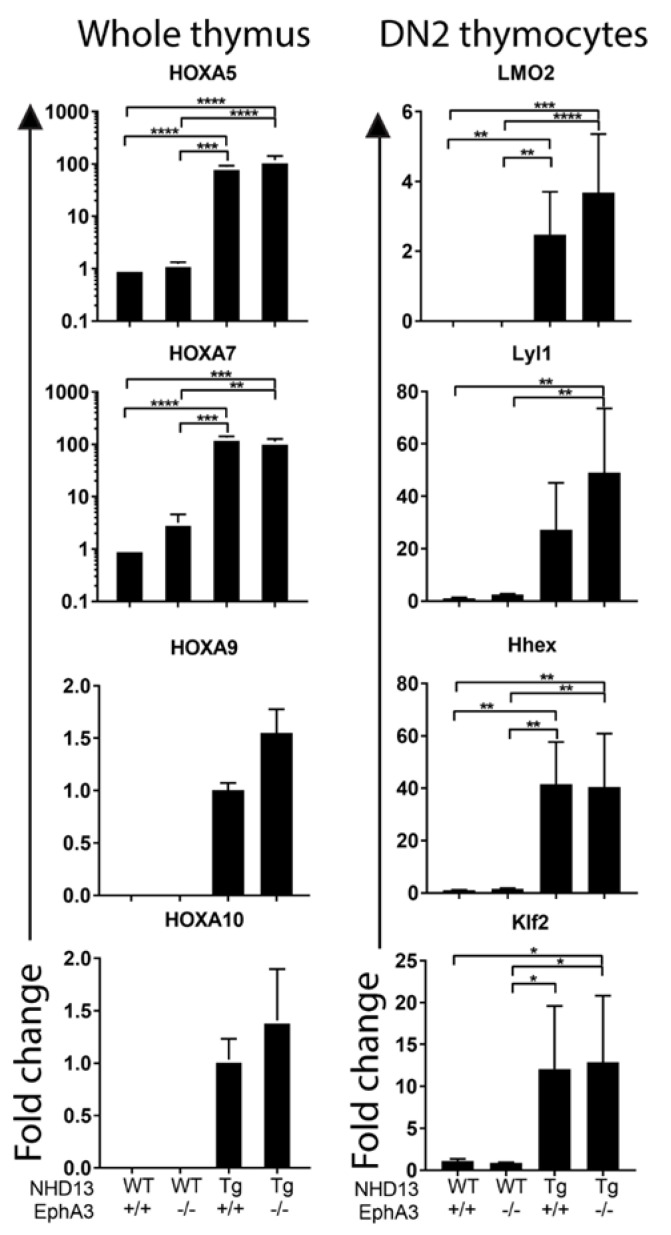
*NHD13* abnormal self-renewal gene expression is unaffected by *EphA3* deletion. RT-PCR testing was performed using 6-week-old whole thymus (left column) or FACS-sorted DN2 (DN2A and DN2B, CD44^+^, CD25^+^ right column) thymocyte samples for each indicated genotype. The average fold change expression of *HoxA5, HoxA7, HoxA9, HoxA10*, *LMO2, Lyl1, Hhex*, and *Klf2* genes was as indicated. Bars represent the mean ± S.E.M of three mice per group. *p*-values were calculated using Student’s *t*-test (* *p* < 0.05, ** *p* < 0.005, *** *p* < 0.0005, **** *p* < 0.00005).

**Figure 5 cancers-13-03858-f005:**
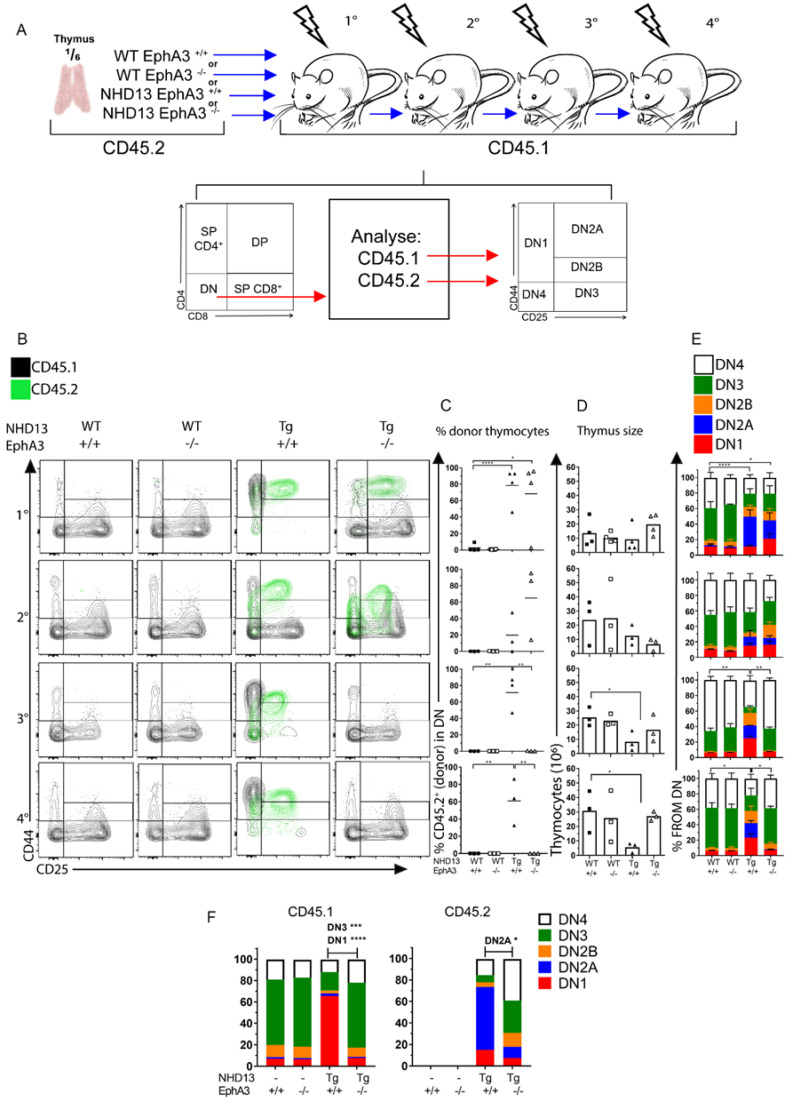
*EphA3* prevents the normal differentiation of wild-type thymocytes in recipients, and mediates the long-term self-renewal capacity of *NHD13* thymocytes upon serial transplantation. (**A**) Schematic representation of experimental design using 6-week-old donor CD45.2 thymuses from WT-*EphA3*^+/+^, WT-*EphA3*^−^^/^^−^, NHD13-*EphA3*^+/+^, or NHD13-*EphA3*^−^^/^^−^ mice. CD45.2 thymocytes equivalent to a ⅙ of a thymus were IV injected into sub-lethally irradiated (6.5 Gy) C57BL/6 CD45.1 recipient mice for primary or further serial transplantations. At 4 weeks post-transplantation, engraftment and contribution of donor CD45.2 cells were analysed via flow cytometry. (**B**) Representative cell profiles of serial transplants (e.g., 1° primary, 2° secondary, 3° tertiary, and 4° quaternary) for each indicated genotype show DN1-4 populations determined by CD44/CD25. Black plots represent recipient CD45.1 cells, while green plots represent engrafted donor CD45.2 cells. (**C**) Contribution of CD45.2 donor thymocytes of each serial transplant. (**D**) Thymus cellularity of each serial transplant. (**E**) Quantitation of DN1–DN4 proportions of each serial transplant. Three biological replicates were performed, and the data represent one biological replicate. (**F**) Quantitation of DN1–DN4 split by CD45.1 or CD45.2. (**C**,**D**) Data represent the mean, points represent 3 individual mice per group, and *p*-values were calculated using Student’s *t*-test (* *p* < 0.05, ** *p* < 0.005, *** *p* < 0.0005, **** *p* < 0.00005). (**E**,**F**) Bars represent the mean ± S.E.M, *p*-values were calculated using a two-way ANOVA, and the significance refers to the DN2A populations.

**Figure 6 cancers-13-03858-f006:**
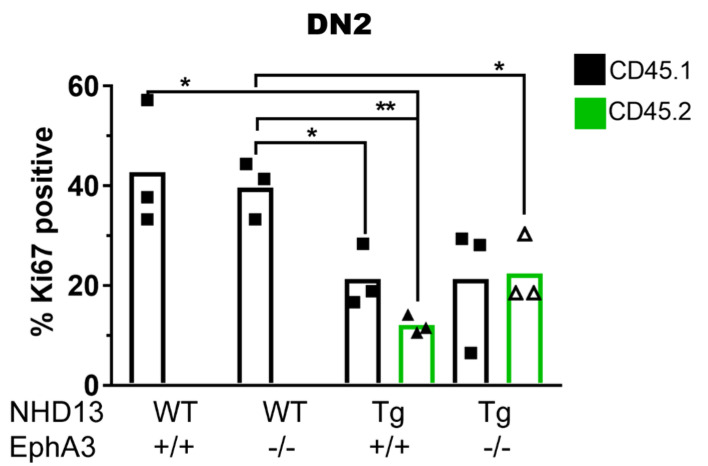
Cell division arrest of WT thymocytes by *NHD13* thymocytes. Cell division analysis of DN2 (DN2A and DN2B) thymocytes in the primary transplant recipient. DN2 (CD44^+^, CD25^+^) recipient (CD45.1) or donor (CD45.2) thymocytes were analysed for cell division/quiescent status using Ki67 and DNA (Hoechst) stains. Bars represent the mean, points represent 3 individual mice per group, and *p*-values were calculated using Student’s *t*-test (* *p* < 0.05, ** *p* < 0.005).

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
