# Peer review of "Cell-Extrinsic Differentiation Block Mediated by EphA3 in Pre-Leukaemic Thymus Contributes to Disease Progression"

_cancers, 2021, doi:10.3390/cancers13153858_

Round 1
Reviewer 1 Report
This study by Pliego Zamora et al describes the role of EphA3 in the NHD13 transgenic mouse by crossing these mice with EphA3 knockout mice showing that EphA3 deletion promotes the differentiation of pre-leukemic thymic T cells and the transplantation of NHD13 thymocytes with EphA3 deletion restores the normal differentiation of the recipient thymic cells.
Overall, this is an interesting study with potentially valuable observations. My main concern is that some of the conclusions in this article may actually be overstatements given that the phenotype of EphA3 deletion is not really prominent.
My major comments are:
- It would be helpful to visualize the EphA3 deletion in the different DN subtypes. The authors show the difference in the expression in the whole DN population but I think it would be helpful to show the EphA3 expression using the gating strategy of Figure 1.
- The authors claim that the EphA3 deletion decreases the incidence of T cell leukemia in the NHD13 model. However, the difference in the incidence 4/25 vs 9/30 is not really convincing.
- The authors report different survival between the EphA3+/+ and EphA3-/- mice. However most mice did not develop T-ALL. If the cause of death was not ALL in the majority of animals, how do they explain the difference in the survival? If they cannot clarify this point, I am not sure how the survival difference fits in the EphA3 concept.
- The authors showed that the cell cycle profile of donor and recipient DN2 thymocytes was not affected by EphA3 deletion. I would expect that there should be some differences noted given the significant differences in the differentiation. Could the authors look at the rest of the thymocyte clusters (i.e. DN1, DN3, DN4)?
- Overall, given the mild phenotype differences, I am not convinced that EphA3 is a critical player affecting the T-ALL biology. For example, given that EphA3 deleted cells appear to lose their stem cell features, it would be interesting to know if they are more sensitive to chemotherapy or other T-ALL treatments (steroids) than EphA3 wild type. In that way the authors could support their statement that EphA3 plays an important role in the T-ALL biology.
Reviewer 2 Report
The authors characterized a mouse model of T cell pre-leukemia NUP98-HOXD13 (NHD13) and described that NHD13 thymocytes overexpress EphA3. EphA3 seems to be essential in mediating the interaction between impaired self-renewal and healthy hematopoietic progenitors.
The manuscript is very elegant and the hypothesis of the existence of an interplay between intrinsic and extrinsic factors modulating the balance between normal and abnormal hematopoiesis is very fascinating.
The manuscript is well written, easy to understand, methodologies are appropriate and conclusions supported by the results.
My minor comments are below:
- Did the authors look at the other EphA family members? Do the author think that any gene dose effect might occur in this model?
- Did the author look at any specific signature of genes related to aging?
Round 2
Reviewer 1 Report
The authors have performed one of the experiments I recommended and have responded at a satisfactory level to my comments.